# Stillbirth rates, service outcomes and costs of implementing NHS England's Saving Babies' Lives care bundle in maternity units in England: A cohort study

Kate Widdows[1], Stephen A. Roberts[2], Elizabeth M. Camacho[3], Alexander E. P. Heazell[1,4]*

**1** Faculty of Biological, Medical and Health, Maternal and Fetal Health Research Centre, School of Medical Sciences, University of Manchester, Manchester Academic Health Science Centre, Manchester, United Kingdom, **2** Centre for Biostatistics, School of Health Sciences, University of Manchester, Manchester Academic Health Science Centre, Manchester, United Kingdom, **3** Manchester Centre for Health Economics, School of Health Sciences, University of Manchester, Manchester Academic Health Science Centre, Manchester, United Kingdom, **4** St. Mary's Hospital, Manchester University NHS Foundation Trust, Manchester Academic Health Science Centre, Manchester, United Kingdom

* alexander.heazell@manchester.ac.uk

**Data Availability Statement:** There is no ethical approval in place to share data from participating Trusts. Anonymised data from staff and patient

## Abstract

### Objective

To assess implementation of the Saving Babies Lives (SBL) Care Bundle, a collection of practice recommendations in four key areas, to reduce stillbirth in England.

### Design

A retrospective cohort study of 463,630 births in 19 NHS Trusts in England using routinely collected electronic data supplemented with case note audit (n = 1,658), and surveys of service users (n = 2,085) and health care professionals (n = 1,064). The primary outcome was stillbirth rate. Outcome rates two years before and after the nominal SBL implementation date were derived as a measure of change over the implementation period. Data were collected on secondary outcomes and process outcomes which reflected implementation of the SBL care bundle.

### Results

The total stillbirth rate, declined from 4.2 to 3.4 per 1,000 births between the two time points (adjusted Relative Risk (aRR) 0.80, 95% Confidence Interval (95% CI) 0.70 to 0.91, P<0.001). There was a contemporaneous increase in induction of labour (aRR 1.20 (95%CI 1.18–1.21), p<0.001) and emergency Caesarean section (aRR 1.10 (95%CI 1.07–1.12), p<0.001). The number of ultrasound scans performed (aRR 1.25 (95%CI 1.21–1.28), p<0.001) and the proportion of small for gestational age infants detected (aRR 1.59 (95%CI 1.32–1.92), p<0.001) also increased. Organisations reporting higher levels of implementation had improvements in process measures in all elements of the care bundle. An economic

questionnaires can be made available subject to appropriate confidentiality agreements between participating organisations. Investigators wishing to obtain data should initially contact the Edgbaston Research Ethics Committee (edgbaston.rec@hra.nhs.uk).

**Funding:** AH, EC and SR recieved funding for this work from NHS England. NHS England approved the research protocol but had no role in study design, data collection and analysis, decision to publish, or preparation of the manuscript.

**Competing interests:** The authors have declared that no competing interests exist.

analysis estimated the cost of implementing the care bundle at ~£140 per birth. However, neither the costs nor changes in outcomes could be definitively attributed to implementation of the SBL care bundle.

## Conclusions

Implementation of the SBL care bundle increased over time in the majority of sites. Implementation was associated with improvements in process outcomes. The reduction in stillbirth rates in participating sites exceeded that reported nationally in the same timeframe. The intervention should be refined to identify women who are most likely to benefit and minimise unwarranted intervention.

## Trial registration

The study was registered on (NCT03231007); www.clinicaltrials.gov.

## Introduction

Stillbirth, defined in the UK as the death of a baby before birth after 24 weeks' gestation [1], has been challenging to reduce, with an annual rate of reduction in the UK of 1.4% between 2000 and 2015, placing the UK in the lowest third of high-income countries (HICs) [2]. Analysis of stillbirth rates within the UK demonstrates significant variation between regions, with the highest stillbirth rates seen in some areas of London, Midlands and the North of England [3,4]. The variation between countries and within the UK suggests that improvement in the stillbirth rate is possible [2]. Given the significant psychological, social and economic impact of stillbirth on mothers and their families [5], further reduction in stillbirth rate in the UK is needed.

Evidence suggests that a significant proportion of these deaths are preventable; Confidential Enquiries into normally formed antepartum stillbirths identified deficiencies in care that contributed to this outcome in 60% of cases, rising to 80% in intrapartum-related deaths [6,7]. Risk factors for stillbirth in HICs include: fetal growth restriction, maternal medical co-morbidities (e.g. diabetes, hypertension), cigarette smoking and maternal perception of reduced fetal movements [8]. Deficiencies in the identification and management of these risk factors has been reported in analyses of stillbirths dating back to 1998 [9]. This information provides a starting point for initiatives to reduce stillbirth in the UK.

To address stillbirth rates in the UK, the Department of Health announced a new ambition to halve the rate of stillbirths by 2030, with a 20% reduction by 2020. In response, NHS England introduced the Saving Babies Lives (SBL) care bundle in 2015 which sought to implement recommendations from established national guidance to address specific risk factors for stillbirth, including i) smoking cessation, ii) fetal growth restriction, iii) reduced fetal movements (RFM) and iv) intrapartum hypoxia [10]. The SBL care bundle consisted of 16 recommendations in four areas of practice: Element 1: Reducing smoking in pregnancy (4 components), Element 2: Risk assessment and surveillance for fetal growth restriction (5 components), Element 3: Raising awareness of reduced fetal movements (2 components) and Element 4: Promoting effective fetal monitoring in labour (5 components). In 2016, NHS England commissioned a retrospective evaluation of the implementation of the SBL care bundle, involving extensive consultation with key stakeholders and participating Trusts a protocol was agreed and published [11]. Here we report the primary and secondary outcomes of the

evaluation, exploratory outcomes where these aid interpretation, and the results of the economic analysis.

## Methods

### Design and study population

We conducted a pragmatic analysis before and after the introduction of the SBL care bundle (the exposure of interest) using longitudinal data collected retrospectively in 19 NHS Trusts in England across 9 clinical networks. These trusts were purposively selected from a larger cohort of maternity units that had participated in the 2015 NHS Tracker Survey to provide a geographically distributed sample throughout England and a range of birth rates and levels of neonatal care and to reflect a range of levels of implementation at the time of selection. Units from London were not included as they were participating in a stepped-wedge cluster trial of implementation of the Growth Assessment Protocol [12]. These trusts were deemed early adopters as they were pilot sites for implementing SBL in 2015 ahead of the national launch in March 2016. For the purposes of this analysis April 2015 was defined as the start of implementation in these 19 Trusts and outcomes were assessed at the nominal dates of April 2013 ("before") and April 2017 ("after") respectively, providing a minimum of two years of data before and after implementation of the SBL care bundle.

Retrospective data collection for the period April 2013 to October 2017 was carried out between September 2017 and January 2018. Recruitment (for surveys) took place between August and December 2017, following approval by the Health Research Authority in June 2017 (Reference 17/WM/0197). The study was registered on www.clinicaltrials.gov (NCT03231007) in July 2017 before commencement of data collection, and the evaluation protocol was published before commencing data analysis [11]. Analysis was performed on anonymized data. The study was reported according to STROBE guidelines [13].

### Variables and data sources

Definitions of data collection methods and outcomes are described in detail in the protocol [11]. In brief, outcome data was collected from routine electronic clinical data held in Trust data repositories where this was available, the data being provided or converted to monthly aggregate totals (e.g. number of births per month). Anonymised postcode data were used to derive the index of multiple deprivation (IMD) and a mean decile computed for each Trust; where this was not provided an estimate was derived as the mean IMD of a surrounding area based on population size and birth numbers.

This was supplemented by clinical audits (20 sequential women per time point and Trust) of growth and birth monitoring practice in unselected women (pre and post implementation), small for gestational age births (pre and post implementation) and women attending hospital with RFM (post-implementation only). Additionally, in the post intervention period women were asked to complete a survey postpartum prior to discharge (informed consent) which included questions on smoking behaviour and interventions received along with experiences of RFM (S1 File). All health professionals involved in delivering maternity care were invited to participate in an online or paper survey regarding their views and experiences of the SBL care bundle (S2 File) if they had been employed in their current Trust prior to the launch of the care bundle initiative in April 2015. Data on the degree of implementation of the SBL care bundle was self-reported by Trusts and implementation scores were derived by applying a score to a Likert scale whether each intervention in each element was implemented: a) all of the time (score 3), b) most of the time (score 2), c) half of the time (score 1), d) not much of the time (score 0), e) never (score 0), or f) not relevant (score 0). These were totalled to give an

implementation score for each element. As there were different numbers of interventions within individual elements, the overall implementation score for each Trust was calculated as the mean of the implementation scores for the four elements, each expressed as a percentage of the maximum score.

## Statistical analysis

Electronic data on stillbirths other clinical outcomes and smoking rates was provided as, or converted to, monthly counts of outcomes and appropriate denominators. Outcomes were fitted using quasi-binomial models to allow for the possibility of overdispersion, except for scan rates, which were fitted using quasi-Poisson models. Generalised linear models with logarithmic link functions were used to estimate a linear trend over time. Trust was included as a covariate thus time trends are estimated on a within-Trust basis. From this model we derived estimates of the outcome rates at dates 2 years either side of the nominal SBL start date of April 2015 and estimated the risk- (or rate) ratio between these two time points as a measure of the change over the implementation period. Models additionally containing a step change at the nominal or reported implementation dates were also considered, but proved to be uninformative as step-changes could not be detected and therefore are not presented.

In order to investigate the relationship with reported implementation status, generalised linear mixed models were fitted with Trust as a random intercept and assessment-date implementation level, Care Level (Tertiary v Secondary), IMD (mean decile) and month as fixed effects. A risk-ratio between no and full implementation is reported.

Audit data and women's questionnaire data were available for one or two time points and the outcomes are presented as proportions of women audited with exact Binomial 95% CI. Where there were data from 2 time points (pre and post implementation), risk ratios between the two time points were derived using binomial regression models with logarithmic link functions. Data analyses were conducted using R (www.R-project.org) [14].

The sample sizes for the various datasets were largely determined pragmatically by the constraints of time and the need to have a reasonable pre and post launch period to assess trends, with an inevitably staggered true implementation. Based on a prevalence of 4.7 normally formed singleton stillbirths per 1,000 total births in 2014 [4], the potential annual number of stillbirths detected was estimated as 470 per 100,000 total births across all study sites. Conservatively, a two year pre versus one year post-comparison was estimated to have 80% power to detect a drop in the primary stillbirth rate from 4.7 to 3.9/1000 –a 17% reduction [11].

## Economic analysis

An economic analysis was carried out using the data generated from the comparisons outlined above alongside additional data reported by the Trusts on the resources they used to implement the SBL care bundle and the level of implementation they reported. These data were also used to estimate what annual costs and stillbirths avoided for the whole of England would be in the scenario that all Trusts implemented all elements of SBL care bundle.

The implementation cost consists of two parts: the direct cost of putting in place each element and the cost of the secondary effects (mode of delivery, induction of labour, ultrasound scans). The direct cost of implementing each element was estimated but it is not possible to determine which element the secondary effects relate to so they have been calculated for the SBL care bundle overall. No additional funding was provided to Trusts to implement the SBL therefore the direct implementation costs should be interpreted as the 'value' of the SBL rather than additional funding required.

Unit costs were derived from published sources [15,16], NHS partners [17], and through email communication with training/software providers. The fees for externally-provided training courses were included in the implementation cost. In-house training was assumed to form part of routine, ongoing continuing professional development (CPD), and so the cost of staff time to complete training was not included in the primary analysis, but was explored in a sensitivity analysis.

The total annual birth rate for England was identified from national published data [18]. The time-series-adjusted stillbirth rates from before and after the implementation date were applied to this number to estimate the difference in the number of stillbirths before and after implementation of the SBL care bundle i.e. the number of stillbirths potentially avoided.

It was necessary to make a number of assumptions in order to estimate the resources and costs associated with the care bundle. As such the costs and outcomes reported should be interpreted as 'best estimates'. The impact of varying some of the assumptions on costs and outcomes were explored in a series of one-way sensitivity analyses.

## Results

The characteristics of the participating Trusts are described in Table 1. The annual birth rate ranged from 2,900 to 8,894. Thirteen out of the 19 Trusts (68%) were secondary care providers with the remainder being tertiary maternity units. The majority (n = 17) of Trusts provided care for women residing in areas between the 2nd and 6th deciles of deprivation. Implementation rates of the SBL care bundle ranged from 0 to 100%, with a mean value of 74%. Response rate on the patient questionnaire was between 9% and 60% of all births.

Table 1. Characteristics of participating trusts.

| Trust | Care Level | Births per year | IMD decile | Implementation Score (%) | Patient Response Rate |
|-------|-----------|-----------------|------------|--------------------------|-----------------------|
| A | Level 2 | 6309 | 3 [2–5] | 100 | 39/424 (9%) |
| B | Level 2 | 4859 | 6 [4–9] | 98 | 34/241 (14%) |
| C | Level 2 | 3970 | 6 [4–8] | 95 | 99/215 (46%) |
| D | Level 2 | 2900 | 3 [2–6] | 93 | 102/297 (34%) |
| E | Level 2 | 3300 | 6 [3–7]* | 78 | 109/1193 (9%) |
| F | Level 3 | 5833 | 6 [4–7] | 90 | 103/357 (29%) |
| G | Level 2 | 3808 | 2 [1–5] | 90 | 105/407 (26%) |
| H | Level 3 | 8550 | 2 [1–5] | 90 | 74/675 (11%) |
| I | Level 2 | 4533 | 7 [5–9]* | 80 | 135/245 (55%) |
| J | Level 2 | 3263 | 5 [4–7] | 80 | 105/193 (54%) |
| K | Level 2 | 1816 | 3 [2–6] | 0 | 120/290 (41%) |
| L | Level 2 | 2962 | 6 [2–9] | 79 | 161/429 (38%) |
| M | Level 2 | 3436 | 4 [2–6] | 75 | 123/205 (60%) |
| N | Level 3 | 4207 | 5 [3–7] | 74 | 103/571 (18%) |
| O | Level 2 | 3205 | 5 [2–7] | 70 | 170/501 (34%) |
| P | Level 3 | 8166 | 8 [6–9] | 56 | 107/213 (50%) |
| Q | Level 3 | 8265 | 2 [1–5] | 61 | 264/820 (32%) |
| R | Level 3 | 8894 | 2 [1–4] | 56 | 105/887 (12%) |
| S | Level 2 | 5240 | 3 [2–6] | 40 | 113/457 (25%) |

Care level (level 2 = secondary, level = 3 tertiary), number of births per year, median and IQR IMD decile [1 = most deprived, 10 = least deprived], SBL implementation at the nominal assessment date and response rate to the patient questionnaire. The overall patient response rate was: 2171/8620 (25%).

*Estimated from geography.

**Table 2. Clinical and service outcomes.**

| | Trusts | Women | Pre | Post | Post v Pre SBL | | Full v no implementation * | |
|---|---|---|---|---|---|---|---|---|
| | | | | | RR | P | RR | P |
| All Stillbirths | 19 | 463,630 | 4.2 | 3.4 | 0.80 (0.70–0.91) | <0.001 | 1.04 (0.73–1.48) | 0.84 |
| Term Singleton Stillbirths | 17 | 387,474 | 1.6 | 1.3 | 0.78 (0.63–0.96) | 0.021 | 1.34 (0.73–2.45) | 0.34 |
| Term Singleton SGA Stillbirths | 17 | 387,474 | 0.6 | 0.4 | 0.69 (0.47–1.02) | 0.060 | 0.80 (0.32–1.98) | 0.63 |
| Preterm births | 15 | 407,484 | 7.4 | 7.9 | 1.06 (1.03–1.10) | <0.001 | 0.91 (0.70–1.18) | 0.47 |
| Preterm Singleton births | 16 | 425,433 | 6.3 | 6.6 | 1.05 (1.02–1.08) | 0.002 | 0.87 (0.64–1.17) | 0.36 |
| NICU admissions | 11 | 282,854 | 3.5 | 4.1 | 1.19 (1.11–1.26) | <0.001 | 0.86 (0.061–12.05) | 0.91 |
| Emergency CS | 15 | 386,817 | 13.7 | 15.0 | 1.10 (1.07–1.12) | <0.001 | 1.02 (0.84–1.24) | 0.85 |
| Elective CS | 17 | 452,944 | 9.9 | 11.8 | 1.19 (1.16–1.23) | <0.001 | 1.01 (0.73–1.38) | 0.97 |
| Induction of labour | 18 | 473,889 | 26.3 | 31.4 | 1.20 (1.18–1.21) | <0.001 | 0.92 (0.73–1.14) | 0.44 |
| Instrumental Births | 18 | 473,889 | 12.2 | 12.4 | 1.01 (0.99–1.04) | 0.25 | 0.83 (0.64–1.08) | 0.17 |
| Spontaneous Births | 18 | 473,889 | 63.4 | 60.4 | 0.95 (0.95–0.96) | <0.001 | 1.00 (0.88–1.15) | 0.94 |
| US scans per pregnancy | 8 | 262,386 | 3.5 | 4.3 | 1.25 (1.21–1.28) | <0.001 | 1.42 (0.94–2.17) | 0.12 |

Columns show the number of Trusts and women providing data, the fitted pre and post implementation rates (±2y from the SBL launch date), the post v pre risk ratio (with 95%CI and significance level) and the risk ratio associated with 100% overall SBL care bundle implementation.
* Adjusted for care level, IMD and calendar month.

Trusts reported that the SBL care bundle was implemented in a phased and gradual manner, with most Trusts implementing some SBL care bundle elements prior to the SBL launch and 12/19 achieving >75% implementation by the end of the study period. Table 1 shows the implementation scores at the assessment date (April 2017) for each Trust with fuller details of the individual elements and time points available in S1 Fig. Many Trusts were unable to give precise dates for the implementation of individual elements of the SBL care bundle. Implementation rates varied between elements with only one Trust stating 100% for all four elements; intrapartum fetal monitoring was the most completely implemented element of the care bundle and screening for SGA infants the least.

## Stillbirth rates

Stillbirth rates declined from 4.2 to 3.4 per 1,000 births over the 4 year period, which was statistically significant (P<0.001), RR of 0.80 (Table 2). Fig 1 shows a steady decline in the rate over time, and we were unable to demonstrate any step changes associated with implementation of the SBL care bundle. Term singleton and stillbirths associated with SGA showed similar decline (Table 2). Pre-term stillbirth showed a slightly smaller decreased from 2.3 per 1,000 births to 1.9 per 1,000 births (Relative Risk 0.82, p = 0.014). We were unable to demonstrate

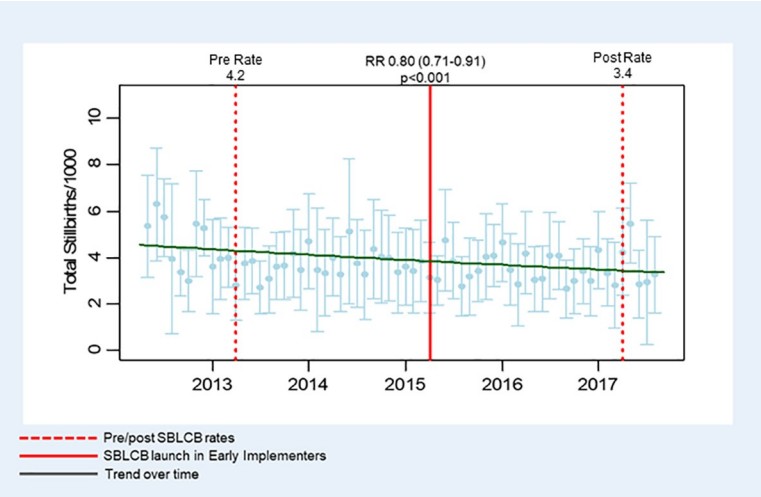

**Fig 1. Average total stillbirth rate across participating sites.** Comparison between pre and post implementation of the Saving Babies Lives care bundle demonstrates a reduction between pre- and post-implementation periods. Republished from Saving Babies' Lives Project Impact and Results Evaluation (SPiRE): A mixed methodology study under a CC BY license, with permission from the University of Manchester, original copyright 2018.

any association between stillbirth rates and the reported level of implementation of the SBL care bundle (Table 2).

## Changes in secondary outcomes

Table 2 summarises the clinical and service outcomes derived from Trust database systems. There were modest increases in the numbers of preterm births, induction of labour and emergency Caesarean sections (CS) over the evaluation period. The number of ultrasound scans per pregnancy increased markedly from 3.5/pregnancy to 4.3/pregnancy in the 8 Trusts providing complete data (S2 Fig). Other Trusts provided partial scan data which was consistent with these trends, but not sufficiently complete for formal analysis. An increase in the number of ultrasound scans performed was associated with a reduction in stillbirth rate (Adjusted Rate Ratio of -0.14 (SE 0.06) stillbirths per additional scan performed P = 0.026). Increases were also seen in elective CS and admissions to NICU, although it must be noted that the quality of data regarding NICU admission was poor. None of these outcomes showed a significant association with implementation of the SBL care bundle.

## Process outcomes

The number of women smoking at delivery declined from 14.3% to 11.8% (P<0.001) over the evaluation period. However, this reflects a decline in the proportion of women smoking at booking rather than an increase in the numbers ceasing smoking during pregnancy (Table 3). 70.1% of women reported being offered a CO test and 99.1% of these women reported accepting the offer, however a smaller proportion of women who smoked were offered referral to smoking cessation services (60.1%) and only 31.0% attended this service. Trusts reporting full implementation of the element had higher proportion of women having a CO test at booking (RR 7.95, Table 3), but no difference in the proportion of women referred for smoking cessation services (RR 1.11, Table 3).

Over the time period of the evaluation, documentation of screening for SGA fetuses improved, with increased documentation of symphysis-fundal height and estimated fetal

**Table 3. Process outcomes.**

| | Source | Trusts | Women | Pre | Post | Post v Pre SBL | | Full v no implementation [*] | |
|---|---|---|---|---|---|---|---|---|---|
| | | | | | | RRpost | P | RR | P |
| **Element 1** | **Smoking Cessation** | | | | | | | | |
| Smoking at Delivery | Women† | 19 | 2,085 | - | 10.1 | - | - | 1.21 (0.67–2.19) | 0.50 |
| Smoking at Delivery | Electronic Data | 14 | 268,736 | 14.3 | 11.8 | 0.83 (0.80–0.86) | <0.001 | 0.91 (0.44–1.86) | 0.80 |
| Offered CO test | Women† | 19 | 1,870 | - | 70.1 | - | - | 7.95 (5.17–12.23) | <0.001 |
| Referred to smoking cessation | Women† | 19 | 311 | - | 60.1 | - | - | 1.11 (0.41–2.98) | 0.82 |
| Ceased Smoking | Women† | 19 | 331 | - | 40.8 | - | - | 0.70 (0.28–1.76) | 0.42 |
| Ceased Smoking | Electronic Data | 11 | 27,550 | 28.7 | 27.0 | 0.94 (0.87–1.01) | 0.098 | 3.99 (0.80–19.99) | 0.11 |
| Smoked in Pregnancy | Women† | 19 | 2,158 | - | 15.7 | - | - | 1.15 (0.71–1.86) | 0.55 |
| Smoking at Booking | Electronic Data | 11 | 203,635 | 16.2 | 14.1 | 0.87 (0.84–0.90) | <0.001 | 0.50 (0.25–1.02) | 0.070 |
| **Element 2** | **Detection and Management of SGA fetus** | | | | | | | | |
| SGA detected | SGA Audit | 17 | 636 | 33.8 | 53.7 | 1.59 (1.32–1.92) | <0.001 | 1.16 (0.34–4.01) | 0.79 |
| Growth chart in records | General Audit | 18 | 720 | - | 97.8 | - | - | 1.21 (0.17–8.80) | 0.84 |
| EFW plotted | SGA Audit | 17 | 636 | 24.7 | 75.7 | 3.06 (2.48–3.76) | <0.001 | 38.24 (7.80–187.46) | <0.001 |
| EFW plotted correctly | SGA Audit | 17 | 582 | 99.2 | 99.1 | 1.00 (0.98–1.01) | 0.80 | Insufficient Data | - |
| SFH plotted | SGA Audit | 17 | 636 | 35.5 | 47.8 | 1.35 (1.11–1.63) | 0.002 | 5.07 (1.42–18.11) | 0.005 |
| **Element 3** | **Fetal Movements** | | | | | | | | |
| RFM leaflet received | Women† | 19 | 1,735 | - | 74.4 | - | - | 2.87 (1.98–4.17) | <0.001 |
| RFM checklist used | RFM Audit | 17 | 339 | - | 52.2 | - | - | 0.44 (0.22–0.88) | 0.010 |
| Attendances for RFM | Women† | 19 | 2,171 | - | 36.5 | - | - | 0.95 (0.70–1.28) | 0.70 |
| Scan after RFM (any) | RFM Audit | 17 | 322 | - | 64.9 | - | - | 0.54 (0.25–1.17) | 0.080 |
| Scan for RFM | Women† | 19 | 793 | - | 29.4 | - | - | 0.85 (0.49–1.46) | 0.52 |
| Heart Monitoring every RFM visit | RFM Audit | 17 | 339 | - | 97.3 | - | - | 0.28 (0.03–2.82) | 0.18 |
| Heart Trace for RFM | Women† | 19 | 793 | - | 73.5 | - | - | 0.81 (0.46–1.43) | 0.42 |
| Induced following RFM | RFM Audit | 17 | 338 | - | 54.7 | - | - | 4.78 (2.29–9.95) | <0.001 |
| **Element 4** | **Fetal Monitoring** | | | | | | | | |
| Buddy and Sticker used | General Audit | 18 | 527 | - | 50.3 | - | - | 33.00 (10.85–100.39) | <0.001 |
| Escalation Protocol used | General Audit | 18 | 224 | - | 98.7 | - | - | Insufficient Data | - |

Columns show the data source, the number of Trusts and women providing data, the fitted pre and* post implementation rates (±2y from the SBL launch date), the post v pre risk ratio (with 95% CI and significance level) and the risk ratio associated with 100% implementation of the relevant SBL element.

[*] Adjusted for care level, IMD and calendar month.

†Women–refers to data obtained via the postnatal questionnaire of women undertaken in the postnatal period.

weight on fetal growth charts. The documentation of these parameters was also significantly higher in Trusts with full implementation of this element (RR 5.07 (symphysis-fundal height), RR 38.24 (estimated fetal weight)). The documented increase in screening for SGA fetuses and in the frequency of ultrasound scan was associated with a 59% increase in the detection rate of SGA fetuses from 33.8% to 53.7% (Table 3). The leaflet regarding RFM was reported to be received by the majority of women (73.1%), and attendance with RFM occurred in 77.3% of episodes of RFM. A lower proportion of cases used a checklist to guide management (52.2%) and management varied from practice recommended in SBL care bundle with 74.4% of women with RFM having a fetal heart rate tracing and 29.4% of women receiving an ultrasound scan. A large proportion of women reporting RFM had an induction of labour (54.7%; Table 3). Women attending trusts with full implementation of the SBL care bundle were more likely to receive the information leaflet about RFM (RR 2.87, Table 3).

We were unable to obtain sufficient data directly from participating Trusts regarding training in interpretation of fetal heart rate traces over the duration of implementation of the SBL care bundle. Data from the staff questionnaires showed that 56.5% of staff (n = 1,064) reported they had received training in the preceding year with 16.8% of those who had been trained reporting they had completed assessment in fetal heart rate interpretation. Staff employed the "buddy system" where at least two professionals interpreted the fetal heart rate trace in 50.2% of audited cases. Trusts with full implementation were more likely to have used the CTG "buddy system" and a standardised recording of the fetal heart rate assessment (RR 33.00).

## Staff views

1,064 health professionals completed the staff survey (details of respondents can be found in S3 File); 78% of respondents were midwives. Overall, 58% (584/905) of respondents were aware of the SBL care bundle, with lowest rates of awareness in junior doctors and ultrasonographers (47% (7/15) and 32% (11/34) of respondents respectively). Almost all staff perceived an increased demand for ultrasound scans (97%, Table 4), rates of induction of labour (98%) and to a lesser extent Caesarean sections (80%). A significant proportion of respondents (41%) perceived that the stillbirth rate had decreased in their unit, although the same proportion perceived it was unchanged (Table 4). Most staff 82% (807/983) agreed with the statement that their unit was actively doing things to improve the safety of mothers and babies.

## Economic analysis

There were 666,025 births (live births and stillbirths) in England in 2016 [18]. Based on data observed from the study sites, the number of women booked is on average 14% higher than the number of births, therefore, it was estimated that there were 759,299 bookings in the same period. The primary estimated cost of implementing the care bundle for the first year in England is £93,116,650, or approximately £140 per birth. The direct cost (excluding secondary costs) is £3,650,357 (4%) of the total cost. The full breakdown of costs are reported in Table 5.

A series of one-way sensitivity analyses were conducted to explore the impact of assumptions made to estimate costs. Selected sensitivity analyses are shown in Table 6 below (see S1 Table for all alternative assumptions explored). The assumption which reduced the estimated cost by the largest amount (-£14,123,076) was that only half of the increase in the number of inductions of labour observed were associated with the SBL care bundle (i.e. an increase of 10% rather than 20%). Including the cost of staff time to complete training courses associated with Elements 2 and 4 of the SBL care bundle increased the estimated cost by the largest amount (+£30,807,840).

As reported above, the stillbirth rate was 0.84/1,000 births lower in the participating Trusts after SBL care bundle was implemented than before. Applying this to the number of births in

**Table 4. Staff views about changes in practice and outcomes in their Trust over the time frame of adoption of the SBL care bundle (n = 1,064).**

| Over the last 5 years | Greatly increased | Slightly increased | Not changed | Slightly decreased | Greatly decreased | Don't know |
|---|---|---|---|---|---|---|
| The demand on ultrasound scanning has... | 839 (87%) | 98 (10%) | 15 (2%) | 2 (0%) | 11 (1%) | 99 |
| The number of stillbirths has... | 10 (1%) | 126 (16%) | 323 (41%) | 260 (33%) | 65 (8%) | 280 |
| The number of babies admitted to a NICU has... | 108 (15%) | 330 (47%) | 214 (30%) | 47 (7%) | 5 (1%) | 360 |
| The number of inductions... | 713 (79%) | 171 (19%) | 17 (2%) | 4 (0%) | 1 (0%) | 158 |
| The number of caesarean sections has... | 270 (31%) | 425 (49%) | 161 (18%) | 13 (1%) | 2 (0%) | 193 |

Percentages calculated as proportion of those who expressed a view regarding the statement.

**Table 5. Estimated annual costs for England associated with the SBL care bundle.**

**A. Direct implementation cost**

| Element | Resources included | Resources excluded | Cost (£) |
|---|---|---|---|
| **Element 1** | i. 9 CO monitors/1000 births (£165 each) <br> ii. D-pieces for monitors, to be replace monthly (£3 each) <br> iii. Mouthpieces for monitors, one per each woman booked (£0.25 each) | i. 10 minutes of midwife time to speak to women who smoke (9–24% of women in study) about smoking cessation and/or do referral <br> ii. Calibration of monitors | £1,394,713 |
| **Element 2*** | i. GAP software set-up (£500/Trust) <br> ii. GAP annual software cost (£1500–5000 depending on size of Trust) | i. Staff time (midwives and sonographers) to attend training course in GAP software run for free by Perinatal Institute <br> ii. Administrator time to generate customised growth charts | £391,000 |
| **Element 3** | i. Trusts instructed to add logos to leaflet and then photocopy from a master copy, two sides of A4 (£0.10 each) | i. Midwife time to discuss leaflet <br> ii. Midwife time to discuss RFM at subsequent visits <br> iii. Attendances with perceived RFM | £66,605 |
| **Element 4** | i. Online training course in CTG interpretation (£60) completed annually by midwives, consultants, and junior doctors | i. Staff time to complete training course | £1,798,039 |
| **Direct Cost** | £3,650,357 (4% of total cost) | | |

**B. Secondary implementation costs**

| Inductions | Induction rate increased from 26.27 to 31.40 per 100 births, costing £847.15 per induction. | | | | £28,945,817 (31% of total cost) |
|---|---|---|---|---|---|
| **Births** | | **Before** | **After** | **Cost** | £26,754,741 (29% of total cost) |
| | Normal (£1704.50) | 63.42 | 61.94[b] | -£16,802,227 | |
| | EMCS (£4553.41) | 13.69 | 15.01 | £40,033,063 | |
| | Instrumental (£3306.71) | 12.25 | 12.41 | £3,523,905 | |
| **Scans** | Number of scans per woman booked increased from 3.51 to 4.35 (24% increase), costing £52.94 per scan. | | | | £33,765,735 (36% of total cost) |
| **Secondary costs** | £89,466,293 (96% of total cost) | | | | |
| **TOTAL** | £93,116,650 | | | | |

EMCS = emergency Caesarean section; *although the use of GROW software to generate customised growth charts was not specified in the care bundle, only 2 out of 19 Trusts included in the analysis said that they did not use it therefore it was included in the costs.

**Table 6. Alternative assumptions for costs associated with the SBL care bundle (total costs across the whole of England).**

| Alternative assumptions | Change in cost versus base case† | Total cost |
|---|---|---|
| 10% increase in induction rate (versus ~20% observed) | £14,123,076 lower | £78,993,574 |
| No direct implementation costs | £3,650,357 lower | £89,466,293 |
| 50% of maternity units use GAP software (versus 100%) | £195,500 lower | £92,921,150 |
| Include cost for 5% of births to attend antenatal clinic with perceived RFM (£75.15/visit) (versus no visit) | £2,569,296 higher | £95,685,946 |
| Include cost of increased rate of elective Caesarean sections (£3,438.12/ delivery) (versus assume all would have been normal births (£1704.50/ delivery)) | £26,711,019 higher | £119,827,669 |
| Include cost of staff time to complete training courses (Element 2: £9,351,300; Element 4: £21,456,540 | £30,807,840 higher | £123,924,491 |

† The total cost in the base case estimate was £93,116,650.

England in 2015/16 (n = 666,052), an estimated 559 stillbirths per year may have been avoided. Using the primary estimate of the cost of the SBL care bundle (£93,116,650), this equates to £166,577 per stillbirth avoided. However, this figure should be interpreted with caution because it is not possible to determine the proportion of either the increase in secondary resource use or the reduction in stillbirths that are directly attributable to the SBL care bundle.

## Discussion

This evaluation of the SBL care bundle found that implementation of the elements was variable, but increased from baseline levels in all early adopter sites. Over the same time frame stillbirth rates fell in the participating maternity units by 20% from 4.2 to 3.4 per 1,000 live births. This rate of reduction was greater than seen across the whole of England over the same period which reduced from 4.2 to 3.7 per 1,000 births from 2013–2017 (4.2 to 3.8 per 1,000 births in units not included in this evaluation) [19]. Data from MBRRACE from participating units suggests that the decrease in intrapartum stillbirths was particularly dramatic, falling from 0.32 per 1,000 births in 2013 to 0.06 per 1,000 births in 2016 (RR 0.18, 95% CI 0.08, 0.45) [3,20]. There were also important changes in secondary outcomes including a reduction in spontaneous vaginal births, increase in preterm births, Caesarean sections, ultrasound scans, and admissions to the neonatal unit. Due to the variation in implementation strategies, phasing and timing we were not able to identify a "step" in our time series analysis which would have suggested a direct association between implementation of the care bundle and the observed reduction in stillbirth or changes in secondary outcomes. However, the degree of implementation was related to important process measures including plotting of fetal growth and assessment of intrapartum fetal heart rate monitoring.

### Strengths and limitations

This evaluation was strengthened by the adherence to a published protocol which used stillbirth as its primary outcome rather than a surrogate measure as often used in this area [11]. The evaluation was conducted in 19 different maternity units reflecting the diversity of UK maternity units and included approximately 14% of the births in the UK. The data presented here represent the major component of the Saving Babies' Lives Project Impact and Results Evaluation (SPiRE project) which also included analysis of clinical practice guidelines and service-user and staff views to provide a holistic view of the impact of implementing the SBL care bundle. The results of the other components are reported elsewhere [21].

This study is limited by it's before-and after study design which was employed pragmatically as the care bundle and its implementation were planned before the evaluation project was commissioned. The degree of implementation was self-reported by participating trusts which may not have accurately reflected the degree of implementation in the clinical service. Furthermore, the variation in levels, phasing, and strategies of implementation meant that there was no discrete moment within or across Trusts where a change in practice could be determined. The use of routinely collected data was both a strength and a limitation of the study. The use of routinely collected data allowed the project to achieve the necessary large sample size and to be undertaken in a comparatively short-time frame, but the use of these data were frequently challenging due to variation in codes and coding strategies employed by individual Trusts. Furthermore, routine data may include recognised coding errors in maternity health episode statistics [22], but as this would affect both the before and after time periods in this analysis the impact of this may be minimal. In some cases, this meant that data on certain items were excluded for some or all Trusts e.g. admission to neonatal unit and rate of hypoxic-ischaemic

encephalopathy. This also meant that it was not possible to include costs (or potential cost-savings) associated with these items in the economic evaluation.

## Meaning of the study: Possible explanations and implications for clinicians and policymakers

Reducing stillbirths (alongside reducing maternal deaths, neonatal deaths, and brain injuries) is an explicit priority of the UK government, with a target to reduce such events by 50% by 2030 and by 20% by 2020. Achieving this aim will require an annual rate of reduction for stillbirth of 3.3%, significantly greater than the 1.4% reported between 2000–15 [2]. If maintained, the annual rate of reduction described in the study sites (~5%) would be sufficient to achieve this ambitious target and represents the most rapid reduction in stillbirth rates in over 20 years and is on target to achieve the national ambition to halve the rate.

In common with earlier national care bundles in perinatal care (e.g. Necrotising Enterocolitis Care Bundle) this evaluation has demonstrated clear improvements in process outcomes, but it was more challenging to identify changes in element-specific clinical outcomes and relate these to the degree of implementation [23]. Although this evaluation was not able to determine whether implementation of the SBL care bundle or any individual element was associated with the observed reduction in stillbirth, the association is plausible. Detection of SGA infants, the strongest risk factor for stillbirth in the UK [24,25], improved by 59% and use of ultrasound scanning, the principal screening method for SGA infants increased by 24%. Importantly, increased frequency of ultrasound scanning was inversely associated with stillbirth, suggesting that a pathway of increased detection of SGA may reduce stillbirth. Unfortunately, causes of stillbirth, particularly fetal growth restriction, are poorly recorded in routinely collected data which prevents analysis to determine whether there has been a reduction in stillbirths from fetal growth restriction [26].

## Unanswered questions and future research

Any argument proposing a link between implementation of SBL and the outcome measures must be weighed against the change in outcomes which were not associated with the SBL care bundle. For example, the 20% increase in elective Caesarean section, which is unlikely to relate to implementation of the SBL care bundle as it contains no recommendations about Caesarean section, is more likely to reflect changes in guidance from the National Institute of Health and Care Excellence published in 2011 [27]. In addition to changes in national policy, we were not able to identify specific initiatives which may have been implemented in participating Trusts contemporaneously. As the SBL care bundle continues to be rolled-out in UK maternity units further evaluation needs to be planned to identify potential causal effects on stillbirth and other clinical outcomes.

Stillbirths are relatively rare events in the UK therefore large, good-quality, datasets are critical to informing efforts to reduce the stillbirth rate. Health policy makers should consider the evaluation of healthcare programmes prior to their implementation, in order that the most robust evaluation possible can be incorporated to ascertain its effects on process and clinical outcomes. The AFFIRM study, which prospectively evaluated increased maternal awareness and staff education about RFM (i.e. Element 3 of the SBL care bundle) [28], demonstrated how a stepped-wedge cluster design can be used effectively in this context.

The analysis presented here provides information about the average effect of implementation of the care bundle across the 19 Trusts. However, we were not able to explore the reasons for the variation in levels of implementation between sites, thus further research is needed to explore the origins of variation seen in different settings. Analysis of clinical practice guidelines

relating to the SBL care bundle in participating maternity units found wide variation in their quality [21]. An ethnographic study of a UK maternity unit with a strong safety record (which was not part of this evaluation) described six mechanisms in place to enhance safety including: clearly articulated, constantly reinforced standards of practice, behaviour and ethics and structural influences on mechanisms for safety (e.g. staffing levels and financial infrastructure) [29]. Thus, we hypothesise that influences accounting for variable implementation may include: human factors, leadership, availability of up to date, accurate clinical practice guidelines and sufficient resource to implement them. These merit exploration in future analyses and the information can then be used to inform future quality improvement programmes.

While the resources required to implement the SBL care bundle may seem large, they should be viewed from a national perspective. In 2014/15 approximately £2.5bn was spent in the NHS on maternity services (with 664,399 births) and in 2016/17 obstetric claims handled by the NHS Litigation Authority accounted for £4.3bn of new claims reported. Based on figures for 2014/15, the maternity tariff per birth was approximately £3,760; the direct costs associated with SBL care bundle are estimated to account for £5 of this with secondary costs estimated at £135 per birth. As the majority of Trusts participating in the study reported using GROW software to produce customised growth charts the costs to do so were included in the estimated costs, however it should be noted that this software was not specified in the SBL care bundle and that alternatives could be used. It was not possible to collect data regarding some important resources relevant to the implementation of SBL care bundle, for example the impact of incorporating additional elements into routine antenatal appointments without increasing the duration of appointments, or the costs associated with NICU admissions, and so it is acknowledged that there is uncertainty around the estimated costs. Future economic analysis of the impact of the SBL care bundle or similar initiatives should also consider the health and social care costs of stillbirth (estimated as £13.1M/year in 2018) [30], which are greater than for a live birth and also extend into pregnancies after stillbirth [31].

## Conclusion

Prior to the launch of the SBL care bundle the UK stillbirth rate was 24th out of 49 HICs and the annual rate of reduction was in the lowest third. Although adoption of the SBL care bundle was variable, all sites implemented elements to some degree. In the same time frame, process outcomes improved and stillbirth rates in the early adopter units fell faster than national rates. However, changes in care had important secondary effects particularly with regard to increased obstetric intervention and consequent resource requirements. This assessment has provided an important insight into the need for evaluation to be planned at the same time as the implementation of large-scale initiatives to ensure that the opportunity to collect data to inform the ongoing development of care is maximised.

## Supporting information

**S1 Checklist. STROBE statement—checklist of items that should be included in reports of *cohort studies*.**
(DOCX)

**S1 Fig. Implementation of elements of the Saving Babies Lives care bundle.** A) Average implementation of all elements at different time frames of the project. B) Level of implementation of each elements at the post-implementation assessment. The total element scores vary due to different number of components in each Element. Element 1: Reducing smoking in pregnancy (4 components), Element 2: Risk assessment and surveillance for fetal growth

restriction (5 components), Element 3: Raising awareness of reduced fetal movements (2 components) and Element 4: Promoting effective fetal monitoring in labour (5 components) Red = <50% of element implemented, Orange = 51–75% of element implemented, Green = >75% of element implemented. Republished from Saving Babies' Lives Project Impact and Results Evaluation (SPiRE): A mixed methodology study under a CC BY license, with permission from the University of Manchester, original copyright 2018.
(TIF)

**S2 Fig. Average total number of ultrasound scans performed per pregnancy.** Across participating sites pre and post implementation of the Saving Babies Lives care bundle demonstrating a 24% increase in the number of ultrasound scans performed. Republished from Saving Babies' Lives Project Impact and Results Evaluation (SPiRE): A mixed methodology study under a CC BY license, with permission from the University of Manchester, original copyright 2018.
(TIF)

**S1 Table. Sensitivity analysis and alternative costs for health economic analysis.**
(DOCX)

**S1 File. Questionnaire for women to complete after giving birth.**
(PDF)

**S2 File. Questionnaire regarding staff views and experiences of the SBL care bundle.**
(PDF)

**S3 File. Characteristics of participants in the questionnaire for health care professionals.**
(PDF)

## Acknowledgments

Holly Reid who participated in the study design. Participants, research midwives and members of professional bodies and charity organisations who participated in the stakeholder groups (Tommy's, RCM, PI, BAPM, RCOG, Sands, Tamba, MAMAs, BMFMS). The authors would like to thank staff from the participating NHS Trusts for providing data and undertaking audits: Barnsley Hospital NHS Foundation Trust, Birmingham Women's and Children's NHS Foundation Trust, Countess of Chester Hospital NHS Foundation Trust, Doncaster and Bassetlaw Hospitals NHS Foundation Trust, Gateshead Health NHS Foundation Trust, Liverpool Women's NHS Foundation Trust, Manchester University NHS Foundation Trust, Norfolk and Norwich University Hospitals NHS Trust, North Cumbria University Hospitals NHS Trust, Oxford University Hospitals NHS Trust, Plymouth Hospital NHS Trust, Royal United Hospitals Bath NHS Foundation Trust, Sherwood Forest Hospitals NHS Foundation Trust, St Helens and Knowsley Teaching Hospitals NHS Trust, Taunton and Somerset NHS Foundation Trust, Mid Yorkshire Hospitals NHS Trust, Royal Devon & Exeter NHS Foundation Trust, University Hospitals of Morecambe Bay NHS Foundation Trust and York Teaching Hospital NHS Foundation Trust.

## Author Contributions

**Conceptualization:** Stephen A. Roberts, Alexander E. P. Heazell.

**Data curation:** Kate Widdows.

**Formal analysis:** Kate Widdows, Stephen A. Roberts, Elizabeth M. Camacho, Alexander E. P. Heazell.

**Funding acquisition:** Stephen A. Roberts, Elizabeth M. Camacho, Alexander E. P. Heazell.

**Methodology:** Elizabeth M. Camacho.

**Project administration:** Kate Widdows, Alexander E. P. Heazell.

**Writing – original draft:** Kate Widdows, Stephen A. Roberts, Elizabeth M. Camacho, Alexander E. P. Heazell.

**Writing – review & editing:** Kate Widdows, Stephen A. Roberts, Elizabeth M. Camacho, Alexander E. P. Heazell.

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
