## [Decision Letter · Decision Letter 0]

28 Oct 2020

PONE-D-20-24240

Stillbirth rates, service outcomes and costs of implementing NHS England’s Saving Babies’ Lives care bundle in maternity units in England: a cohort study

PLOS ONE

Dear Dr. Heazell,

Thank you for submitting your manuscript to PLOS ONE. After careful consideration, we feel that it has merit but does not fully meet PLOS ONE’s publication criteria as it currently stands. Therefore, we invite you to submit a revised version of the manuscript that addresses the points raised during the review process.

We look forward to receiving your revised manuscript.

Kind regards,

Ju Lee Oei

Academic Editor

PLOS ONE

Journal Requirements:

2. In ethics statement in the manuscript and in the online submission form, please provide additional information about the patient records/samples used in your retrospective study. Specifically, please ensure that you have discussed whether all data/samples were fully anonymized before you accessed them and/or whether the IRB or ethics committee waived the requirement for informed consent. If patients provided informed written consent to have data/samples from their medical records used in research, please include this information.

3. In your Methods section, please provide additional information about the participant recruitment method and the demographic details of your participants. Please ensure you have provided sufficient details to replicate the analyses such as: a) the recruitment date range (month and year), b) a description of any inclusion/exclusion criteria that were applied to participant recruitment, c) a table of relevant demographic details, d) a statement as to whether your sample can be considered representative of a larger population, e) a description of how participants were recruited, and f) descriptions of where participants were recruited and where the research took place.

4. Please include additional information regarding the survey or questionnaire used in the study and ensure that you have provided sufficient details that others could replicate the analyses. For instance, if you developed a questionnaire as part of this study and it is not under a copyright more restrictive than CC-BY, please include a copy, in both the original language and English, as Supporting Information.

5. We noted in your submission details that a portion of your manuscript may have been presented or published elsewhere.

"This manuscript is original, but a portion of these data were presented previously in an archived report of the SPiRE evaluation which is available here –

https://www.manchester.ac.uk/discover/news/download/573936/evaluationoftheimplementationofthesavingbabieslivescarebundleinearlyadopternhstrustsinenglandjuly2018-2.pdf). "

Please clarify whether this publication was peer-reviewed and formally published. If this work was previously peer-reviewed and published, in the cover letter please provide the reason that this work does not constitute dual publication and should be included in the current manuscript.

6. We note that you have indicated that data from this study are available upon request. PLOS only allows data to be available upon request if there are legal or ethical restrictions on sharing data publicly. For information on unacceptable data access restrictions, please see http://journals.plos.org/plosone/s/data-availability#loc-unacceptable-data-access-restrictions.

7. Your ethics statement should only appear in the Methods section of your manuscript. If your ethics statement is written in any section besides the Methods, please delete it from any other section.

Reviewers' comments:

Reviewer's Responses to Questions

**Comments to the Author**

1. Is the manuscript technically sound, and do the data support the conclusions?

Reviewer #1: Yes

Reviewer #2: Yes

2. Has the statistical analysis been performed appropriately and rigorously? 

Reviewer #1: Yes

Reviewer #2: Yes

3. Have the authors made all data underlying the findings in their manuscript fully available?

Reviewer #1: No

Reviewer #2: Yes

4. Is the manuscript presented in an intelligible fashion and written in standard English?

Reviewer #1: Yes

Reviewer #2: Yes

5. Review Comments to the Author

Reviewer #1: Manuscript: PONE-D-20-24240

Stillbirth rates, service outcomes and costs of implementing NHS England’s Saving Babies’ Lives care bundle in maternity units in England: a cohort study

General comments

In this manuscript the authors describe a cohort study used to perform and in-depth evaluation of a national initiative to address stillbirth rates in the UK. Although necessarily pragmatic in design, this is a well-constructed and reported study. The reported finding of a reduction in stillbirth rates across participating sites that exceeds that reported nationally in the same timeframe will be of interest to a broad, international readership.

Specific comments

Abstract

• Line 42, p 3- ‘Implementation of the SBL care bundle increased over time in the majority of sites with associated improvements in process outcomes. This statement lacks clarity and needs re- wording.

Methods

• Line 114, p6- where authors describe method for scoring degree of implementation. What do you mean by totalled to give an implementation score for each element? Did each element have multiple components that were each scored then added or was there only one degree of implementation question per element? Was there any weighting given to certain components? i.e. some more important than others? It would be helpful for the reader to have clearly stated somewhere what ‘implementing’ element 1, 2, 3, 4, refers to (even though this may have been previously published).

• Line 162, p9- Was there a specific reason staff time to complete training was not included in costs, especially for externally-provided training? Why were cost associated with neonatal outcomes not included? eg admission to NICU

Discussion

• Line 267, p14- Authors mention a decrease in SB rate for ‘whole of England’, how much of this decrease can be accounted for by the trusts included in this evaluation? It may be worthwhile comparing SB rate reduction for ‘early adopters’ and ‘all other’ sites as separate groups as well as total ‘whole of England’ reduction?

Limitations

• Could the authors provide comment on whether self-reported degree of implementation as described is a limitation of the study and how this may have contributed to challenges if any with interpretation of results.

General

• Are the authors able to offer any further insights as to why adoption of the bundle was variable across sites and what unit characteristics made a ‘high degree’ implementer? E.g. influence of leadership, resources, other?

• It would be of value to understand maternity staff attitudes towards the ‘bundle approach’, satisfaction with the way it was implemented and how this might have influenced process outcomes. Were there additional results from the survey of staff that are not reported here that might describe this?

• Consider replacing the word “ patients” with “ women” and “ deliveries” with “births” throughout.

Tables and Figures

• Table 3. – In source column consider replacing ‘Patient’ with ‘women”

• Supplementary Figure 1- Heading check grammar- ‘implantaion’? Could the authors explain for section B why the x-axis has a different scale for each element? This goes back to previous comment in methods, it is unclear how you are calculating the score for each element. Could you also please explain why site K has an overall score of 0% yet does have scores for each element?

• Given element 4 is intrapartum monitoring it would be helpful to see SB rate broken down into antepartum vs intrapartum to further explain changes in SB rate due to implementation of element 4.

Reviewer #2: Thanks for allowing me to review your manuscript. I have several comments and questions.

1. Can you provide more details about the process for selecting the 19 NHS trusts involved in this evaluation? How was it that a trust with an implementation score of 0 was selected?

2. I found the economic analysis difficult to follow due to the number of assumptions made. In particular, it seems that all of the increased interventions are costed as 'additional costs' whereas it may also be that increases in certain interventions such as induction of labour and caesarean section actually save money in terms of reducing other adverse outcomes. I do wonder if the economic analysis is worthy of a separate paper, which would then allow a much more comprehensive review of the costs of this bundle implementation.

3. The data presented gives rates of 'total stillbirths' and 'term stillbirths' but it would also be of interest to see how rates changed at lower gestations. It might be expected that the interventions would have a major impact on late gestation stillbirth but it appears that there have been come changes at earklier gestations as well. Can you provide that information? I am sure it would be of interst to the readers.

6. PLOS authors have the option to publish the peer review history of their article (what does this mean?). If published, this will include your full peer review and any attached files.

Reviewer #1: No

Reviewer #2: **Yes: **David Ellwood

---

## [Author Response · Author response to Decision Letter 0]

3 Feb 2021

 The manuscript has been formatted to match PLOS ONE’s style requirements. 

2. In ethics statement in the manuscript and in the online submission form, please provide additional information about the patient records/samples used in your retrospective study. Specifically, please ensure that you have discussed whether all data/samples were fully anonymized before you accessed them and/or whether the IRB or ethics committee waived the requirement for informed consent. If patients provided informed written consent to have data/samples from their medical records used in research, please include this information.

All data were fully anonymized before analysis. Participants in the online survey were asked to indicate their consent to participation in their response. 

3. In your Methods section, please provide additional information about the participant recruitment method and the demographic details of your participants. Please ensure you have provided sufficient details to replicate the analyses such as: a) the recruitment date range (month and year), b) a description of any inclusion/exclusion criteria that were applied to participant recruitment, c) a table of relevant demographic details, d) a statement as to whether your sample can be considered representative of a larger population, e) a description of how participants were recruited, and f) descriptions of where participants were recruited and where the research took place.

The Methods section includes the date ranges for specific components of the study. Table 1 describes the characteristics of the maternity units (NHS Trusts) that participated in the study. No demographic data for individual participants were collected. We have described how participating units were selected and where the research took place.

4. Please include additional information regarding the survey or questionnaire used in the study and ensure that you have provided sufficient details that others could replicate the analyses. For instance, if you developed a questionnaire as part of this study and it is not under a copyright more restrictive than CC-BY, please include a copy, in both the original language and English, as Supporting Information.

The questionnaire for women who gave birth at participating Trusts and the questionnaire for staff working at participating trusts have been included as supplementary files.

5. We noted in your submission details that a portion of your manuscript may have been presented or published elsewhere.

"This manuscript is original, but a portion of these data were presented previously in an archived report of the SPiRE evaluation which is available here –

https://www.manchester.ac.uk/discover/news/download/573936/evaluationoftheimplementationofthesavingbabieslivescarebundleinearlyadopternhstrustsinenglandjuly2018-2.pdf). "

Please clarify whether this publication was peer-reviewed and formally published. If this work was previously peer-reviewed and published, in the cover letter please provide the reason that this work does not constitute dual publication and should be included in the current manuscript.

This work was published on the University of Manchester website and a limited number of printed copies but was not peer-reviewed. 

6. We note that you have indicated that data from this study are available upon request. PLOS only allows data to be available upon request if there are legal or ethical restrictions on sharing data publicly. For information on unacceptable data access restrictions, please see http://journals.plos.org/plosone/s/data-availability#loc-unacceptable-data-access-restrictions.

We did not ask participating units whether we could share the data they provided to the study. Therefore, we are not able to share it legally or under the terms of the research ethics approval granted for this study. Data requests should be sent to the Chief Investigator who can then approach the research ethics committee to determine whether access can be facilitated. 

7. Your ethics statement should only appear in the Methods section of your manuscript. If your ethics statement is written in any section besides the Methods, please delete it from any other section.

Reviewer #1:

General comments

In this manuscript the authors describe a cohort study used to perform and in-depth evaluation of a national initiative to address stillbirth rates in the UK. Although necessarily pragmatic in design, this is a well-constructed and reported study. The reported finding of a reduction in stillbirth rates across participating sites that exceeds that reported nationally in the same timeframe will be of interest to a broad, international readership.

Specific comments

Abstract

• Line 42, p 3- ‘Implementation of the SBL care bundle increased over time in the majority of sites with associated improvements in process outcomes. This statement lacks clarity and needs re- wording.

This statement has been re-worded and now reads “Implementation of the SBL care bundle increased over time in the majority of sites. Implementation was associated with improvements in process outcomes.” (lines 42-44)

Methods

• Line 114, p6- where authors describe method for scoring degree of implementation. What do you mean by totalled to give an implementation score for each element? Did each element have multiple components that were each scored then added or was there only one degree of implementation question per element? Was there any weighting given to certain components? i.e. some more important than others? It would be helpful for the reader to have clearly stated somewhere what ‘implementing’ element 1, 2, 3, 4, refers to (even though this may have been previously published).

The section on the implementation scoring has been revised for clarity. Units were asked to report a score for the implementation for each component of each element (ranked 0-3). These were then added to give the total for each element which is presented in Supplementary Figure 1. As some elements had different numbers of components the overall implementation was computed as the mean of each element expressed as a percentage of the total score (now edited lines 122-125). Thus all components were equally weighted within each element, and each element equally weighted in the overall implementation score. The authors have edited the manuscript to clarify what element 1, 2, 3 and 4 refers to in the introduction section (Lines 71-75) and also added the word “element” to caption of sup figure 1 to make this more explicit.

• Line 162, p9- Was there a specific reason staff time to complete training was not included in costs, especially for externally-provided training? Why were cost associated with neonatal outcomes not included? E.g. admission to NICU.

The time to complete training was not included in the primary estimate of costs due to the assumption that in-house training would be part of continuing professional development for staff. However, this has now been included in the sensitivity analyses reported in Table 6. Data on neonatal admissions were not included in the evaluation because of significant inconsistencies between data supplied from different providers which also meant that the associated costs could not be included in the economic analysis. The authors recognise that this is a limitation of the study (lines 335-336). 

Discussion

• Line 267, p14- Authors mention a decrease in SB rate for ‘whole of England’, how much of this decrease can be accounted for by the trusts included in this evaluation? It may be worthwhile comparing SB rate reduction for ‘early adopters’ and ‘all other’ sites as separate groups as well as total ‘whole of England’ reduction?

The stillbirth rate for the whole of England across the duration of this study (as derived from central MBRRACE data) was 4.22 per 1,000 births in 2013 to 3.74 per 1,000 births in 2017. In participating units the rates fell from 4.23 per 1,000 births in 2013 to 3.40 per 1,000 births in 2017, in comparison units not participating in this evaluation the rates were 4.21 per 1,000 births in 2013 and 3.8 per 1,000 births in 2017. We have included this information in the discussion section. However, there were a number of sites not included in this evaluation that were implementing aspects of the SBL care bundle and one site (site S) within the evaluation who had very limited implementation. Therefore, we do not think it is appropriate to view this as a comparison between early adopters and all other sites. 

Limitations

• Could the authors provide comment on whether self-reported degree of implementation as described is a limitation of the study and how this may have contributed to challenges if any with interpretation of results.

The authors agree with the reviewer that self-reported degree of implementation is a limitation of the study and this has been added to the limitations section of the study (lines 323-325). However, the process measures do give us an objective measure of implementation.

General

• Are the authors able to offer any further insights as to why adoption of the bundle was variable across sites and what unit characteristics made a ‘high degree’ implementer? E.g. influence of leadership, resources, other? 

Unfortunately we did not collect information about the leadership structure or resources of individual Trusts to be able to make a valid judgement about whether these factors affected the degree of implementation of the care bundle across sites. We collected some information from staff about their perceptions of safety in the maternity unit where they work. This has been added to the manuscript. We have added this as an area for future work in the discussion section of the manuscript (lines 379-381). 

• It would be of value to understand maternity staff attitudes towards the ‘bundle approach’, satisfaction with the way it was implemented and how this might have influenced process outcomes. Were there additional results from the survey of staff that are not reported here that might describe this? 

We collected data from staff (n=1,064) regarding their views about the care bundle. We have previously reported staff views about clinical practice guidelines in their units (Lau et al. BMJ Open Quality 2020;9:e000756) and have not repeated these findings here. We have included information about staff views about the care bundle including the demand for ultrasound scanning, the stillbirth rate, induction rate and number of Caesarean sections in the results section. We have also included staff views about safety in their maternity unit (lines 261-267). 

• Consider replacing the word “patients” with “women” and “deliveries” with “births” throughout.

Patients have been replaced with women and deliveries have been replaced with births throughout the manuscript.

Tables and Figures

• Table 3. – In source column consider replacing ‘Patient’ with ‘women”

Patient has been replaced by women in the source column of Table 3. 

• Supplementary Figure 1- Heading check grammar- ‘implantaion’? Could the authors explain for section B why the x-axis has a different scale for each element? This goes back to previous comment in methods, it is unclear how you are calculating the score for each element. Could you also please explain why site K has an overall score of 0% yet does have scores for each element?

The heading has been altered to read “implementation”. An explanation has been added to explain why the x-axis is different for each element in the methods section. This has been reiterated in the figure legend. In figure A) implementation at baseline, implementation date and post-implementation is shown. In Figure B, the breakdown for the different elements is given at the post-implementation time point is shown. This is stated in the legend. The x-axis for the charts in section A has been altered to read % Implementation score. 

• Given element 4 is intrapartum monitoring it would be helpful to see SB rate broken down into antepartum vs intrapartum to further explain changes in SB rate due to implementation of element 4.

Unfortunately we did not collect information on the timing of stillbirth so are not able to present the intrapartum stillbirth rate in our manuscript. However, the statistics from MBRRACE demonstrate that the intrapartum stillbirth rate fell from 0.32 per 1,000 births in participating units in 2013 to 0.06 per 1,000 births in 2016 (Relative risk 0.18, 95% CI 0.08, 0.45). As these data were not obtained from our data we have not included this information in the results section but have added it to the discussion section of the manuscript (lines 303-305). 

Reviewer #2:

Thanks for allowing me to review your manuscript. I have several comments and questions.

1. Can you provide more details about the process for selecting the 19 NHS trusts involved in this evaluation? How was it that a trust with an implementation score of 0 was selected?

All Trusts that were deemed “early adopters” of the SBLCB in 2015 were eligible to take part; these were sites that completed the 2015 NHS England Tracker Survey indicating that they were implementing the SBLCB. Initially, Trusts were selected to take part in the evaluation to compare outcomes in providers reporting full, partial or low implementation stages as reported in the Tracker Survey, But at the time of the study in 2016 most Trusts had increased their implementation so the original study design of full, partial, low was not possibleUnits were purposively sampled to provide a geographically distributed sample throughout England and a mixture of Trusts with different numbers of births and levels of neonatal care. Units from London were not included as they were participating in a stepped-wedge cluster trial of implementation of the Growth Assessment Protocol (Trials 2019 Mar 4;20(1):154. doi: 10.1186/s13063-019-3242-6). The Trust (S) had an implementation score of 0 at baseline which improved to 40% post-implementation, so although the implementation from this site was far from that anticipated, some aspects of the SBL care bundle had been introduced. 

Information regarding the selection of sites has been included in the methods section (lines 88-92). 

2. I found the economic analysis difficult to follow due to the number of assumptions made. In particular, it seems that all of the increased interventions are costed as 'additional costs' whereas it may also be that increases in certain interventions such as induction of labour and caesarean section actually save money in terms of reducing other adverse outcomes. I do wonder if the economic analysis is worthy of a separate paper, which would then allow a much more comprehensive review of the costs of this bundle implementation.

We have rephrased the reporting of the results of the economic analysis to make it easier to follow. Due to the pragmatic and post hoc nature of the evaluation it was necessary to make a number of assumptions in order to estimate the costs associated with the care bundle as it was not possible to collect additional data. This also means that a more comprehensive review of the costs is not possible and so we feel that this combined paper is appropriate. We agree that there may have been some downstream cost-savings associated with fewer adverse outcomes, for example if there were fewer NICU admissions, however data were not available to quantify these impacts and so it was not possible to include them in this analysis. This has been added as a limitation in the discussion (lines 335-336). 

3. The data presented gives rates of 'total stillbirths' and 'term stillbirths' but it would also be of interest to see how rates changed at lower gestations. It might be expected that the interventions would have a major impact on late gestation stillbirth but it appears that there have been come changes at earlier gestations as well. Can you provide that information? I am sure it would be of interest to the readers.

We do not have sufficiently detailed data to calculate the stillbirth rate for individual gestations or groups of gestational ages, but we are able to divide total stillbirths into “term stillbirths” and “preterm stillbirths”. The rate of term stillbirth decreased from 1.9 per 1,000 births pre-intervention to 1.4 per 1,000 births post-intervention (Relative Risk 0.77, p=0.005) and the rate of pre-term stillbirth decreased from 2.3 per 1,000 births to 1.9 per 1,000 births (Relative Risk 0.82, p=0.014). We have added this information to the manuscript to emphasise that there were reductions in both term and preterm stillbirths (lines 213-215).

---

## [Decision Letter · Decision Letter 1]

9 Mar 2021

PONE-D-20-24240R1

Stillbirth rates, service outcomes and costs of implementing NHS England’s Saving Babies’ Lives care bundle in maternity units in England: a cohort study

PLOS ONE

Dear Dr. Heazell,

Thank you for submitting your manuscript to PLOS ONE. After careful consideration, we feel that it has merit but does not fully meet PLOS ONE’s publication criteria as it currently stands. Therefore, we invite you to submit a revised version of the manuscript that addresses the points raised during the review process, including reviewer 3 who has requests for clarification of reporting, which I think can be answered easily. 

We look forward to receiving your revised manuscript.

Kind regards,

Ju Lee Oei

Academic Editor

PLOS ONE

Journal Requirements:

Reviewers' comments:

Reviewer's Responses to Questions

**Comments to the Author**

1. If the authors have adequately addressed your comments raised in a previous round of review and you feel that this manuscript is now acceptable for publication, you may indicate that here to bypass the “Comments to the Author” section, enter your conflict of interest statement in the “Confidential to Editor” section, and submit your "Accept" recommendation.

Reviewer #2: All comments have been addressed

Reviewer #3: (No Response)

2. Is the manuscript technically sound, and do the data support the conclusions?

Reviewer #2: Yes

Reviewer #3: Partly

3. Has the statistical analysis been performed appropriately and rigorously? 

Reviewer #2: Yes

Reviewer #3: No

4. Have the authors made all data underlying the findings in their manuscript fully available?

Reviewer #2: Yes

Reviewer #3: No

5. Is the manuscript presented in an intelligible fashion and written in standard English?

Reviewer #2: Yes

Reviewer #3: Yes

6. Review Comments to the Author

Reviewer #2: (No Response)

Reviewer #3: This is a well-conducted study; however its reporting does not follow the appropriate reporting guidelines, and I am unable to make out a number of key features; for example:

- it is unclear whether the investigators obtained individual/participant level data or whether the data was collected already aggregated at trust or hospital hospital-level. I understand that it could have been a mix of these for different variables e.g. individual-level outcomes and trust-level covariates, however, even this needs to be made clear for the various variables when they are described.

- the number of units studied should be reported very clearly when indicating the size of this study. For example, the abstract describes this as a retrospective cohort study of 463,630 births in 19 trusts and mentions 1,658 audits, an surveys of 2085 users and 1,064 health professionals, but it is still not quite clear which of these units the outcomes reported pertain to; for example, did you obtain stillbirth events for each particpant or did you obtain them aggregated at hospital/trust level? It is also not clear how the numbers of units break down over the various times the assessments relate to.

- the key variables for this analysis are not clearly described; for example, what were the outcomes and how were they defined/measured and at what level (individual? aggregated at higher level? repeated over time?); what was the main exposure variable and how was it defined? any covariates and how they were measured/defined for the analysis?

- the approach to analysis is not clearly described. Without knowing what the outcome and explanatory variables were, or the units of analysis (individual?/trust?/hospital?), it is not possible to assess whether the models described were suitable i.e. appropriate for the outcome variables, level of aggregation, any hierarchical relationships in the data, assessing change etc. For instance, the RRs in Table 2 compare post- and pre- periods (however defined), but it is not clear whether the rates in those periods are obtained from a model on the outcomes of individual women, or a model on aggregated outcomes at hospital level across 19 trusts, or even aggregated outcomes at trust level. Furthermore, the results tables should present number of units of observation contributing to the pre- and the post- rates/proportions for each outcome, not just the overall totals as currently presented. Ideally, for the reporting of outcomes, assuming they are all rates, you should report the number of units (e.g. individuals), the number of events and the rate in the pre- and in the post- period, followed by the rate ratio, confidence intervals and p-value for the test of comparison of the two periods. Crude and adjusted rate ratios/CIs/p-values should be presented if unadjusted and adjusted models were used. For all outcomes, please be clear about how they are aggregated; for example when the change in the number of women smoking is reported as 14.3% to 11.8%, was this pooled across all women in the two periods or was this first aggregated at hospital or trust level and then pooled and compared? The answer to this question is important because it makes a difference in how the variability of the outcome is determined and whether this is indeed a statistically significant difference or not (this also applies to how the rate ratios are calculated all through).

Overall, I would recommend a clearer overall reporting (especially of the methods and results) to be consistent with the reporting guidelines appropriate for this study, and a more detailed description of the statistical methods to enable assessment of their suitability and potential replication by authors of similar studies.

7. PLOS authors have the option to publish the peer review history of their article (what does this mean?). If published, this will include your full peer review and any attached files.

Reviewer #2: No

Reviewer #3: No

---

## [Author Response · Author response to Decision Letter 1]

26 Mar 2021

We would like to thank the reviewers for their comments on our manuscript. We note that Reviewer#2 has no further comments on our manuscript. 

Reviewer #3: This is a well-conducted study; however its reporting does not follow the appropriate reporting guidelines, and I am unable to make out a number of key features; for example:

The study report was written according to the STROBE checklist. This was already uploaded as a supporting document to the PLoSOne website. We have included this information in the manuscript to ensure that readers are aware that we followed reporting guidelines (line 102).

- it is unclear whether the investigators obtained individual/participant level data or whether the data was collected already aggregated at trust or hospital hospital-level. I understand that it could have been a mix of these for different variables e.g. individual-level outcomes and trust-level covariates, however, even this needs to be made clear for the various variables when they are described.

The reviewer is correct that we used a mix of different methods of data collection for individual components of the analysis. We described these in the published protocol all these details are available to readers. We state in the methods the three different data structures

- Implementation data at Trust level (i.e. already aggregated data) (line 122)

- Clinical outcomes and smoking as monthly counts per centre (line 107ff)

- Audit data at specified time points (lines 111-114)

- Post-partum patient questionnaire (line 115 - 116). We have now clarified that this individual questionnaire data was only obtained in the post-intervention period (line 114).

Table 3 gives the source of data for each outcome. We have now made it explicit that “women” in table 3 refers to the questionnaire.

- the number of units studied should be reported very clearly when indicating the size of this study. For example, the abstract describes this as a retrospective cohort study of 463,630 births in 19 trusts and mentions 1,658 audits, an surveys of 2085 users and 1,064 health professionals, but it is still not quite clear which of these units the outcomes reported pertain to; for example, did you obtain stillbirth events for each participant or did you obtain them aggregated at hospital/trust level? It is also not clear how the numbers of units break down over the various times the assessments relate to.

Full details of the sources of each data are provided in the published protocol. The information on the numbers of units and women providing data for each outcome is provided in tables 2 and 3. The survey of women and questionnaires from health professionals were obtained from all 19 participating units. Data regarding the characteristics of staff participants has been added as a supplementary file (S3). The data regarding stillbirths were obtained and then aggregated at unit level. 

Line 107 states: “outcome data was collected from routine electronic clinical data held in Trust data repositories… monthly aggregate totals”. 

- the key variables for this analysis are not clearly described; for example, what were the outcomes and how were they defined/measured and at what level (individual? aggregated at higher level? repeated over time?); what was the main exposure variable and how was it defined? any covariates and how they were measured/defined for the analysis?

The protocol provided detailed information on the data sources and is publicly available and cited in the manuscript. It is clearly stated that all outcome data is aggregated to site/month, although for practical purposes some variables were aggregated by the analysis team rather than the data provider. The main exposure variable is the SBL intervention as stated in the title and the first line of the method section (lines 83-84). No covariates are mentioned as there were none, other than the Trust (line 137)

- the approach to analysis is not clearly described. Without knowing what the outcome and explanatory variables were, or the units of analysis (individual?/trust?/hospital?), it is not possible to assess whether the models described were suitable i.e. appropriate for the outcome variables, level of aggregation, any hierarchical relationships in the data, assessing change etc. For instance, the RRs in Table 2 compare post- and pre- periods (however defined), but it is not clear whether the rates in those periods are obtained from a model on the outcomes of individual women, or a model on aggregated outcomes at hospital level across 19 trusts, or even aggregated outcomes at trust level. 

We argue that the analytical approach is clear in the manuscript. The main analyses are clearly stated to be based on data aggregated at the trust/month level (lines 134-135). The estimates are defined clearly as estimates 2 years before and 2 years post the nominal implementation date (Line 138) from a linear trend model (line 137). For example, the data supplied by the participating units included the number of stillbirths, number of births etc. so in some cases data were aggregated by the team.

Furthermore, the results tables should present number of units of observation contributing to the pre- and the post- rates/proportions for each outcome, not just the overall totals as currently presented. Ideally, for the reporting of outcomes, assuming they are all rates, you should report the number of units (e.g. individuals), the number of events and the rate in the pre- and in the post- period, followed by the rate ratio, confidence intervals and p-value for the test of comparison of the two periods. 

Although the data analysed were obtained from individual units the relevant number is the number of women in the model (i.e. the totals that are presented), and the RR presented is based on the slope over time – converted to a pre-post RR for ease of interpretation. 

Crude and adjusted rate ratios/CIs/p-values should be presented if unadjusted and adjusted models were used. 

There were no adjusted models for the primary analyses (other than the inclusion of Trust). The secondary analyses by implementation status did include covariates, but as this was a secondary analysis we believe that adding unadjusted results would confuse rather than enhance the manuscript. This was done as a sensitivity analysis and made no substantive difference to the result.

For all outcomes, please be clear about how they are aggregated; for example when the change in the number of women smoking is reported as 14.3% to 11.8%, was this pooled across all women in the two periods or was this first aggregated at hospital or trust level and then pooled and compared? The answer to this question is important because it makes a difference in how the variability of the outcome is determined and whether this is indeed a statistically significant difference or not (this also applies to how the rate ratios are calculated all through).

Smoking data is (see table 3) derived from two sources. The Electronic data is aggregated on the per trust per month basis as described above. The women’s questionnaire data was analysed in the same way as the audit data (This has been clarified by adding “and women’s questionnaire data” to line 145). With specific regard to the reviewer’s comment on the change in frequency of smoking, 14.3% is the point estimate 2 years prior to the implementation of the SBL care bundle and 11.8% is at the assessment point 2 years after the implementation. These data were obtained by calculating the pooled proportion of women who smoked aggregated from the individual unit data.

Overall, I would recommend a clearer overall reporting (especially of the methods and results) to be consistent with the reporting guidelines appropriate for this study, and a more detailed description of the statistical methods to enable assessment of their suitability and potential replication by authors of similar studies

We believe all the information requested was available in brief in the manuscript and in detail in the published protocol. A few details have been clarified in the manuscript where these were missing. We have adhered to the STROBE reporting guidleines.

---

## [Decision Letter · Decision Letter 2]

1 Apr 2021

Stillbirth rates, service outcomes and costs of implementing NHS England’s Saving Babies’ Lives care bundle in maternity units in England: a cohort study

PONE-D-20-24240R2

Dear Dr. Heazell,

We’re pleased to inform you that your manuscript has been judged scientifically suitable for publication and will be formally accepted for publication once it meets all outstanding technical requirements.

Kind regards,

Ju Lee Oei

Academic Editor

PLOS ONE

Additional Editor Comments (optional):

Reviewers' comments:

Reviewer's Responses to Questions

**Comments to the Author**

1. If the authors have adequately addressed your comments raised in a previous round of review and you feel that this manuscript is now acceptable for publication, you may indicate that here to bypass the “Comments to the Author” section, enter your conflict of interest statement in the “Confidential to Editor” section, and submit your "Accept" recommendation.

Reviewer #3: All comments have been addressed

2. Is the manuscript technically sound, and do the data support the conclusions?

Reviewer #3: (No Response)

3. Has the statistical analysis been performed appropriately and rigorously? 

Reviewer #3: (No Response)

4. Have the authors made all data underlying the findings in their manuscript fully available?

Reviewer #3: (No Response)

5. Is the manuscript presented in an intelligible fashion and written in standard English?

Reviewer #3: (No Response)

6. Review Comments to the Author

Reviewer #3: (No Response)

7. PLOS authors have the option to publish the peer review history of their article (what does this mean?). If published, this will include your full peer review and any attached files.

Reviewer #3: No

---

## [Editor Report · Acceptance letter]

7 Apr 2021

PONE-D-20-24240R2 

Stillbirth rates, service outcomes and costs of implementing NHS England’s Saving Babies’ Lives care bundle in maternity units in England: a cohort study 

Dear Dr. Heazell:

I'm pleased to inform you that your manuscript has been deemed suitable for publication in PLOS ONE. Congratulations! Your manuscript is now with our production department. 

Kind regards, 

on behalf of

Dr. Ju Lee Oei 

Academic Editor

PLOS ONE